# Stochastic Diffusion: A Diffusion Based Model for Stochastic Time Series Forecasting

## Abstract

Recent successes in diffusion probabilistic models have demonstrated their strength in modelling and generating different types of data, paving the way for their application in generative time series forecasting. However, most existing diffusion based approaches rely on sequential models and unimodal latent variables to capture global dependencies and model entire observable data, resulting in difficulties when it comes to highly stochastic time series data. In this paper, we propose a novel **Stoch**astic **Diff**usion (StochDiff) model that integrates the diffusion process into time series modelling stage and utilizes the representational power of the stochastic latent spaces to capture the variability of the stochastic time series data. Specifically, the model applies diffusion module at each time step within the sequential framework and learns a step-wise, data-driven prior for generative diffusion process. These features enable the model to effectively capture complex temporal dynamics and the multi-modal nature of the highly stochastic time series data. Through extensive experiments on real-world datasets, we demonstrate the effectiveness of our proposed model for probabilistic time series forecasting, particularly in scenarios with high stochasticity. Additionally, with a real-world surgical use case, we highlight the model's potential in a medical application.

## 1 Introduction

The recent advancements of diffusion probabilistic models has demonstrated their superior abilities in capturing complex data distributions and generating high quality samples across different types of data including image, text, and audio (Ramesh et al., 2021; Kong et al., 2021; Rombach et al., 2022; Saharia et al., 2022; Ruiz et al., 2023). This strength of diffusion based models in data modelling and generation has led to their wide applications in generative time series forecasting where models extract temporal dynamics from historical data and approximate the distribution of future values based on learned features and auxiliary information.(Rasul et al., 2021; Shen and Kwok, 2023; Alcaraz and Strodthoff, 2023; Fan et al., 2024; Li et al., 2024).

Existing diffusion based time series forecasting models predominantly employ sequential architectures such as Recurrent Neural Networks (RNN) (Rasul et al., 2021; Fan et al., 2024), Transformers (Li et al., 2024), or a State Space Model (Alcaraz and Strodthoff, 2023) to first capture temporal dynamics from historical sub-series and learn the underlying patterns that can be used for predicting future values. Once the temporal features are extracted, diffusion modules are applied to generate future time series based on these features in a **post-hoc** fashion, with little integration between the temporal modelling and the diffusion process. Although these models have achieved success with various homogeneous time series data where relationships between the historical and future data are mostly consistent, they still face certain challenges: (1) For heterogeneous time series data like clinical patient data, high variability between each individual can introduce extra stochasticity and make it challenging to model the historical data. (2) The latent variable of the diffusion models is commonly sampled from a unimodal distribution, which exhibits limited capability to encode the full temporal dynamics and uncertainty of real-world time series data.

To address these challenges, we propose a novel diffusion based time series forecasting model called **Stoch**astic **Diff**usion (StochDiff). Unlike existing approaches that apply diffusion process only after time series modelling, StochDiff directly integrates the **diffusion process** into the **modelling stage** to capture the intrinsic characteristics and multi-modal behaviors of complex data, rather than relying

on a post-hoc diffusion process. Specifically, the diffusion module is applied at each **time step** of **sequential framework**, enabling the model to effectively learn both temporal dependencies and stochasticity at every step of time series. This step-wise application of diffusion process allows the latent variable to encode the specific distribution of individual values at each time step, making it more effective for the model to learn the individualized, complex patterns in highly stochastic time series data. Additionally, an auxiliary variable is learned to complement the latent variable, forming the prior distribution of conditional generation in reverse diffusion process. This variable, referred to as **data-driven prior**, is learned from both temporal dynamics extracted by sequential framework and the data at each time step. This prior enables the diffusion module to capture intrinsic characteristics and the multi-modal behavior of the complex data, which is beneficial for modelling highly stochastic time series. Finally, a Gaussian Mixture Model is fitted on sampled future data to obtain more accurate point-wise forecasting results during the inference stage.

We conduct extensive experiments on six real-world datasets across various domains. Results show that StochDiff can achieve competitive performance on homogeneous time series data and superior performance on heterogeneous time series data compared to existing state-of-the-art diffusion based time series forecasting models ($\sim 15\%$ and $\sim 14\%$ improvements in predictive accuracy on two clinical datasets). Furthermore, a case study using real-world surgical data further highlights the model's effectiveness in handling individual variability and efficiency in multi-step prediction, demonstrating its potential in a medical application.

Our contributions can be summarized as follows:

- To the best of our knowledge, our proposed StochDiff is the first diffusion based model that integrates diffusion process into time series modelling stage, with a step-wise, data-driven prior designed specifically for stochastic time series data modelling.
- Through extensive experiments on real-world datasets, we demonstrate the competitive performance of StochDiff, especially on highly stochastic time series data.
- Using an intra-operative data from cochlear implant surgery, we showcase the model's effectiveness in handling individual variability and its efficiency in making multi-step prediction, highlighting its potential for a real-world medical application.

## 2 RELATED WORKS

The Diffusion Probabilistic Models (DPM) is firstly proposed by Sohl-Dickstein et al. (2015), where a forward and reverse diffusion process is used to create a prior distribution and transform it back to complex data distribution by leveraging the thermodynamic analogy. Building on DPM, the Denoising Diffusion Probabilistic Model (DDPM) (Ho et al., 2020) refines these ideas into more practical and effective generative modelling approach using neural networks. This has paved the way for the widespread research into diffusion models and their applications in generating various types of data such as image, audio, and text (Ramesh et al., 2021; Kong et al., 2021; Rombach et al., 2022; Ramesh et al., 2022; Saharia et al., 2022; Ruiz et al., 2023; Zhang et al., 2023).

The strength of diffusion models in data modelling and generation has led to their applications in time series forecasting tasks. The very first work is TimeGrad (Rasul et al., 2021) which combines a diffusion module with RNN to autoregressively generate future time series data. TimeDiff (Shen and Kwok, 2023), on the other hand, applies the non-autoregressive approach and introduces a future mix-up strategy to enhance the time series forecasting. MG-TSD (Fan et al., 2024) applies TimeGrad on different granularity of time series data to improve the model in long-period time series forecasting tasks. Instead of Recurrent networks, Transformer and state space models are also utilized to capture the temporal dynamics of time series data. TMDM (Li et al., 2024) focuses on Transformer based forecasting models, and improves their performance with the diffusion generative module. SSSD (Alcaraz and Strodthoff, 2023) develops structured state space model and combines with diffusion model to improve the accuracy for time series forecasting tasks. Besides time series forecasting, diffusion models have also been applied to other time series analysis tasks including imputation and generation (Tashiro et al., 2021; Kollovieh et al., 2024).

The existing diffusion based time series forecasting models typically apply the the diffusion process **after** the entire observed time series is processed by sequential modules, such as RNN in TimeGrad. In these models, the sequential modules first extract temporal features from observed time series,

then the diffusion modules focus on generating the future time series based on the learned temporal features. In contrast, our proposed model integrates the **diffusion process** directly into **modelling stage** by applying diffusion module at each time step of sequential model, improving its ability to model highly stochastic time series data and forecast more accurate and diverse future values.

## 3 PRELIMINARIES

We provide some necessary background and terminologies. To avoid any confusion, the **diffusion steps** are denoted as superscript $^n$ and **time steps** of time series data are denoted as subscript $_t$.

### 3.1 DENOISING DIFFUSION PROBABILISTIC MODEL

Diffusion probabilistic models generate a data distribution with a series of latent variables:

$$p_\theta(\boldsymbol{x}^0, \boldsymbol{x}^1, \ldots, \boldsymbol{x}^N) = p_\theta(\boldsymbol{x}^{0:N}) := p(\boldsymbol{x}^N) \prod_{n=1}^{N} p_\theta(\boldsymbol{x}^{n-1}|\boldsymbol{x}^n) \tag{1}$$

where, $\boldsymbol{x}^0$ is the data variable and $\boldsymbol{x}^{1:N}$ are latent variables of the same dimension as the data variables. By integrating over the latent variables we can get the data distribution as $p_\theta(\boldsymbol{x}^0) := \int p_\theta(\boldsymbol{x}^{0:N}) d\boldsymbol{x}^{1:N}$ (Sohl-Dickstein et al., 2015).

To construct these latent variables, Diffusion Models use a *forward diffusion process* that gradually adds Gaussian noise into the data and finally converts data into standard Gaussian noise over $N$ scheduled steps: $q(\boldsymbol{x}^{1:N}|\boldsymbol{x}^0) := \prod_{n=1}^{N} q(\boldsymbol{x}^n|\boldsymbol{x}^{n-1}), q(\boldsymbol{x}^n|\boldsymbol{x}^{n-1}) := \mathcal{N}(\boldsymbol{x}^n; \sqrt{1-\beta_n}\boldsymbol{x}^{n-1}, \beta_n \boldsymbol{I})$. Where, $\beta_n$ are learnable parameters representing the scheduled variance level over $N$ steps. In practice, we often fix $\beta_n$ to small positive constants to simplify the model.

Based on the mathematical property of forward diffusion, we can directly sample $\boldsymbol{x}^n$ at an arbitrary step $n$ from $\boldsymbol{x}^0$ using the close form expression:

$$q(\boldsymbol{x}^n|\boldsymbol{x}^0) = \mathcal{N}(\boldsymbol{x}^n; \sqrt{1-\overline{\alpha}_n}\boldsymbol{x}^0, (1-\overline{\alpha}_n)\boldsymbol{I}) \tag{2}$$

where, $\alpha^n := 1 - \beta_n$ and $\overline{\alpha}_n := \prod_{i=1}^{n} \alpha_i$.

Once the latent variables are created, a *reverse diffusion process* in the form of equation 1 will learn to remove noise from each $\boldsymbol{x}^n$ to reconstruct the original data $\boldsymbol{x}^0$. The start point $p(\boldsymbol{x}^N)$ is commonly fixed as $\mathcal{N}(\boldsymbol{x}^N; 0, \boldsymbol{I})$, and each subsequent transition is given by the following parametrization:

$$p_\theta(\boldsymbol{x}^{n-1}|\boldsymbol{x}^n) := \mathcal{N}(\boldsymbol{x}^{n-1}; \boldsymbol{\mu}_\theta(\boldsymbol{x}^n, n), \boldsymbol{\delta}_\theta(\boldsymbol{x}^n, n)). \tag{3}$$

Training the diffusion model is accomplished by uniformly sampling $n$ from $\{1, \ldots, N\}$ and optimizing the Kullback-Leibler (KL) divergence between the posteriors of forward and reverse process:

$$\mathcal{L}_n = D_{KL}(q(\boldsymbol{x}^{n-1}|\boldsymbol{x}^n)||p_\theta(\boldsymbol{x}^{n-1}|\boldsymbol{x}^n)). \tag{4}$$

Although $q(\boldsymbol{x}^{n-1}|\boldsymbol{x}^n)$ is unknown, it can be tractable when conditioned on $\boldsymbol{x}^0$ given equation 2: $q(\boldsymbol{x}^{n-1}|\boldsymbol{x}^n, \boldsymbol{x}^0) = \mathcal{N}(\boldsymbol{x}^{n-1}; \tilde{\boldsymbol{\mu}}_n(\boldsymbol{x}^n, \boldsymbol{x}^0), \tilde{\beta}_n \boldsymbol{I})$. Where, $\tilde{\boldsymbol{\mu}}_n(\boldsymbol{x}^n, \boldsymbol{x}^0) := \frac{\sqrt{\overline{\alpha}_{n-1}}\beta_n}{1-\overline{\alpha}_n}\boldsymbol{x}^0 + \frac{\sqrt{\alpha_n}(1-\overline{\alpha}_{n-1})}{1-\overline{\alpha}_n}\boldsymbol{x}^n$, $\quad \tilde{\beta}_n := \frac{1-\overline{\alpha}_{n-1}}{1-\overline{\alpha}_n}\beta_n$.

For parametrization in equation 3, normally we follow Ho et al. (2020) to fix $\boldsymbol{\delta}_\theta(\boldsymbol{x}^n, n)$ at $\boldsymbol{\sigma}_n^2 \boldsymbol{I}$ where $\boldsymbol{\sigma}_n^2 = \overline{\beta}_n := \frac{1-\overline{\alpha}_{n-1}}{1-\overline{\alpha}_n}\beta_n, \overline{\beta}_1 = \beta_1$, and use neural networks to model $\boldsymbol{\mu}_\theta(\boldsymbol{x}^n, n)$.

With all the simplifications and transformations, we can rewrite the objective in equation 4 as:

$$\mathcal{L}_n = \frac{1}{2\boldsymbol{\sigma}_n^2}||\tilde{\boldsymbol{\mu}}_n(\boldsymbol{x}^n, \boldsymbol{x}^0, n) - \boldsymbol{\mu}_\theta(\boldsymbol{x}^n, n)||^2. \tag{5}$$

Note that in equation 5, due to the flexibility of neural networks, $\boldsymbol{\mu}_\theta(\boldsymbol{x}^n, n)$ can be modelled in two ways: $\boldsymbol{\mu}(\epsilon_\theta)$ and $\boldsymbol{\mu}(\boldsymbol{x}_\theta)$. The former one is widely used by Ho et al. (2020) and its derivations which

involves training a **noise prediction model** $\epsilon_\theta(\boldsymbol{x}^n, n)$. The later one trains a **data prediction model** $\boldsymbol{x}_\theta(\boldsymbol{x}^n, n)$ to obtain the $\boldsymbol{\mu}(\boldsymbol{x}_\theta)$:

$$\boldsymbol{\mu}(\boldsymbol{x}_\theta) = \frac{\sqrt{\alpha_n}(1 - \overline{\alpha}_{n-1})}{1 - \overline{\alpha}_n}\boldsymbol{x}^n + \frac{\sqrt{\overline{\alpha}_{n-1}}\beta_n}{1 - \overline{\alpha}_k}\boldsymbol{x}_\theta(\boldsymbol{x}^n, n). \qquad (6)$$

and the corresponding objective function is:

$$\mathcal{L}_{\boldsymbol{x}} = \mathbb{E}_{n, \boldsymbol{x}^0}[||\boldsymbol{x}^0 - \boldsymbol{x}_\theta(\boldsymbol{x}^n, n)||^2]. \qquad (7)$$

## 3.2 PROBABILISTIC TIME SERIES FORECASTING

Assume we have a multivariate time series $\boldsymbol{x}_{1:T}^0 = \{\boldsymbol{x}_1^0, \boldsymbol{x}_2^0, \ldots, \boldsymbol{x}_{t_0}^0, \ldots, \boldsymbol{x}_T^0\}$, where, $\boldsymbol{x}_t^0 \in \mathbb{R}^d$ is a $d-$dimensional vector representing $d$ measurements for an event happening at time step $t$. Probabilistic Time Series Forecasting involves: (1) modelling the observed data $\boldsymbol{x}_{1:t_0-1}^0$ and (2) predicting the unseen future data $\boldsymbol{x}_{t_0:T}^0$ by modelling the conditional distribution of future data given the observed data:

$$q_\chi(\boldsymbol{x}_{t_0:T}^0 | \boldsymbol{x}_{1:t_0-1}^0) = \prod_{t=t_0}^T q_\chi(\boldsymbol{x}_t^0 | \boldsymbol{x}_{1:t-1}^0). \qquad (8)$$

## 4 STOCHASTIC DIFFUSION FOR TIME SERIES FORECASTING

In this section, we discuss the details of our proposed **Stoch**astic **Diff**usion (StochDiff) model. An overview of the model structure is provided in Figure 1. The model comprises two parts: Modelling

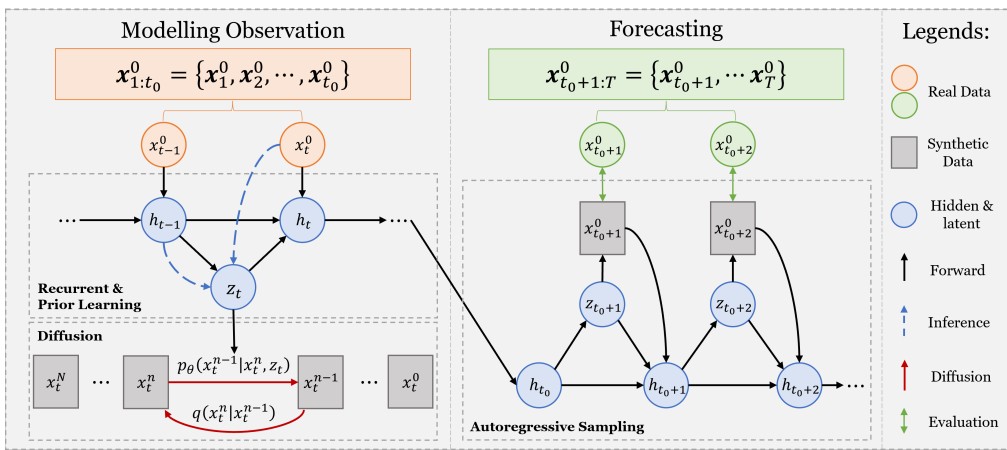

Figure 1: Stochastic Diffusion for Time Series Forecasting

and Forecasting. In the Modelling part, *Recurrent and Prior Learning* firstly learns the prior vector $\boldsymbol{z}_t$: it consists of an *RNN* module that converts the original time series data $\boldsymbol{x}_t^0$ into the hidden units $h_t$, and a *Prior Encoder* that encodes the previous hidden units $h_{t-1}$ into the current prior vector $\boldsymbol{z}_t$. Secondly, the *Diffusion* module integrates the learned prior vector $\boldsymbol{z}_t$ into its latent variables to reconstruct the original data $\boldsymbol{x}_t^0$. The Forecasting part involves autoregressive generation of future data: the last hidden unit $h_{t_0}$ from the observation is taken to learn the first prior vector $\boldsymbol{z}_{t_0+1}$ in the prediction window. The predicted $\boldsymbol{x}_{t_0+1}^0$ is sampled from the diffusion model and is input to the *RNN* module to obtain corresponding hidden unit $h_{t_0+1}$ and the prior vector $z_{t_0+2}$ for the next time step. A graphic decomposition of the modelling operations is provided in Appendix B to help with the understanding of StochDiff.

## 4.1 DATA-DRIVEN PRIOR KNOWLEDGE WITH STEP-WISE MODELLING

In most existing diffusion based time series forecasting models (Rasul et al., 2021; Shen and Kwok, 2023; Fan et al., 2024; Li et al., 2024), the observed time series is modelled entirely as sub-sequences.

This allows the sequential module to extract the intrinsic temporal properties that can be used to predict future dynamics. However, several challenges exist for this strategy: Firstly, when modelling highly stochastic data, such as clinical data which are highly variable between different patients. The model may not be able to learn informative knowledge from historical sub-sequences, leading to failures in forecasting. Additionally, the prior distribution over the latent variable $p(\boldsymbol{x}^N)$ is commonly fixed to be the standard Gaussian distribution $\mathcal{N}(0, \boldsymbol{I})$ in order to simplify the computation and parameter estimation. This unimodal latent variable may introduce inflexibility and inexpressiveness into the data modelling, making it less appropriate to represent the complex temporal systems of highly stochastic and non-linear real-world time series data.

To address these challenges and develop an efficient learning paradigm for highly stochastic time series, we integrate diffusion process into time series modelling stage by applying the diffusion module at each **time step** of **sequential module**, enabling the model to effectively learn both temporal dependencies and stochasticity of data at each time step of time series. To enhance the step-wise learning, we build on previous works (Chung et al., 2015; Aksan and Hilliges, 2019) to design a data-driven prior variable at each time step of time series data:

$$\boldsymbol{z}_t \sim p_z(\boldsymbol{z}_t|\boldsymbol{x}_{1:t-1}^0, \boldsymbol{z}_{1:t-1}) := \mathcal{N}(\hat{\boldsymbol{\mu}}_\theta(h_{t-1}), \hat{\boldsymbol{\delta}}_\theta(h_{t-1})), \quad h_{t-1} = \boldsymbol{f}_\theta(\boldsymbol{x}_{t-1}^0, \boldsymbol{z}_{t-1}, h_{t-2}) \quad (9)$$

where, $\hat{\boldsymbol{\mu}}_\theta(\cdot), \hat{\boldsymbol{\delta}}_\theta(\cdot)$ are functions that simulate the parameters of prior distribution, and can be approximated by neural networks. $h_t$ are the hidden states of the neural network $\boldsymbol{f}_\theta$ which is used to model the time series data.

We use *Long Short Term Memory Networks (LSTM)* as the time series modelling backbone. Hence, $\boldsymbol{f}_\theta$ in equation 9 is *LSTM* and each $h_t$ is the hidden state of it. The *Prior Encoder* includes both $\hat{\boldsymbol{\mu}}_\theta(\cdot), \hat{\boldsymbol{\delta}}_\theta(\cdot)$ that learn the distribution parameter and a *Fully Connected Network (FCN)* that projects the sampled random variable into $\boldsymbol{z}_t$ for the purpose of optimization.

With the dynamic, data-driven prior distribution (equation 9), the latent variables sampled from them are able to (1) encode more informative representations, (2) better approximate the real data distributions, and subsequently, (3) benefit the time series data modelling.

The prior encoder (equation 9) establishes an initial assumption on the distribution of the latent variable $\boldsymbol{z}_t$. According to Bayesian inference, the posterior distribution $p(\boldsymbol{z}_t|\boldsymbol{x}_{1:t}^0, \boldsymbol{z}_{1:t-1})$ is essential to capture the relationship between the latent variables and the observed data. However, since the true posterior is generally intractable, we following existing work to approximate it with another encoder (Kingma and Welling, 2014; Rezende et al., 2014). This encoder, informed by the previous hidden unit $h_{t-1}$ and the current data $\boldsymbol{x}_t$ (blue dash lines in Figure 1), guides the posterior approximation:

$$q_z(\boldsymbol{z}_t|\boldsymbol{x}_{1:t}^0, \boldsymbol{z}_{1:t-1}) := \mathcal{N}(\boldsymbol{\mu}_{\boldsymbol{z},t}(h_{t-1}, \boldsymbol{x}_t^0), \boldsymbol{\delta}_{\boldsymbol{z},t}(h_{t-1}, \boldsymbol{x}_t^0)). \quad (10)$$

Similarly, it also includes both $\boldsymbol{\mu}_{\boldsymbol{z},t}(\cdot), \boldsymbol{\delta}_{\boldsymbol{z},t}(\cdot)$ that learn the distribution parameters and the FCN that projects the learnable $\boldsymbol{z}_t$.

## 4.2 Time Series Forecasting via Conditional Generative Diffusion

With the learned prior distribution $p_z(\boldsymbol{z}_t|\boldsymbol{x}_{1:t-1}^0, \boldsymbol{z}_{1:t-1})$, we can introduce prior knowledge into the diffusion latent variables. Note that, in previous work (Li et al., 2024), prior knowledge is learned from the observed sub-sequences and directly replaces the latent variable for both the forward (at the end point) and reverse (at the start point) diffusion process. However, due to the statistical properties of the forward process, the covariance matrix of the prior vector must be fixed to identity matrix $\boldsymbol{I}$. This introduces extra constraints to the learning of prior knowledge. Thus, in our method, to enable fully learned prior knowledge, we retain the original end point $\boldsymbol{x}_t^N \sim \mathcal{N}(0, \boldsymbol{I})$ for the forward diffusion process, and integrate the prior knowledge into the the reverse diffusion process as a condition. Thus, the parametrization of the reverse diffusion process is now conditioned on latent variable $\boldsymbol{z}_t$:

$$p_\theta(\boldsymbol{x}_t^{n-1}|\boldsymbol{x}_t^n, \boldsymbol{z}_t) = \mathcal{N}(\boldsymbol{x}_t^{n-1}; \boldsymbol{\mu}_\mathcal{C}(\boldsymbol{x}_t^n, n, \boldsymbol{z}_t), \boldsymbol{\delta}_\mathcal{C}(\boldsymbol{x}_t^n, n, \boldsymbol{z}_t)) \quad (11)$$

where $\boldsymbol{\mu}_\mathcal{C}, \boldsymbol{\delta}_\mathcal{C}$ are functions that compute the parameters of the Gaussian distribution, with a fusion function $\mathcal{C}$ integrating $z_t$ into corresponding parameters. $\mathcal{C}$ can be simulated by a neural network which, in our case, is a cross-attention mechanism. Additionally, we also fix the fused covariance matrix $\boldsymbol{\delta}_\mathcal{C}(\boldsymbol{x}_t^n, n, \boldsymbol{z}_t)$ at $\sigma_n^2 \boldsymbol{I}$ for computational simplicity.

Integrating equation 11 into the generative diffusion part (equation 1), and applying the probabilistic time series forecasting formulation (equation 8), we have a variational generative diffusion process for time series forecasting. The details of this formula's integration are provided in Appendix A:

$$p_\theta(\boldsymbol{x}_{t_0:T}^0|\boldsymbol{x}_{1:t_0-1}^0, \boldsymbol{z}_{1:t_0-1}) =$$

$$\int_{\boldsymbol{x}_{t_0:T}^{1:N}} \int_{\boldsymbol{z}_{t_0:T}} \prod_{t=t_0}^{T} p(\boldsymbol{x}_t^N|\boldsymbol{z}_t) \prod_{n=1}^{N} p_\theta(\boldsymbol{x}_t^{n-1}|\boldsymbol{x}_t^n, \boldsymbol{z}_t) p_z(\boldsymbol{z}_t|\boldsymbol{x}_{1:t-1}^0, \boldsymbol{z}_{1:t-1}) d\boldsymbol{z}_{t_0:T} d\boldsymbol{x}_{t_0:T}^{1:N}. \quad (12)$$

Following equation 6, our new data prediction model is now built on $\boldsymbol{x}_t^n$, $n$ and $\boldsymbol{z}_t$. The sampled data at each diffusion step is:

$$\boldsymbol{x}_t^{n-1} = \frac{\sqrt{\alpha_n}(1 - \overline{\alpha}_{n-1})}{1 - \overline{\alpha}_n}\boldsymbol{x}_t^n + \frac{\sqrt{\overline{\alpha}_{n-1}}\beta_n}{1 - \overline{\alpha}_k}\boldsymbol{x}_\theta(\boldsymbol{x}_t^n, n, \boldsymbol{z}_t). \quad (13)$$

### 4.3 DUAL OBJECTIVE OPTIMIZATION

The objective of the proposed StochDiff is the combination of: prior knowledge learning and the denoising diffusion process at each time step. The prior knowledge learning is associated with the variational lower bound (Kingma and Welling, 2014) that optimizes the approximate posterior $q_z(\boldsymbol{z}_t|\boldsymbol{x}_{1:t}^0, \boldsymbol{z}_{1:t-1})$ by minimizing the KL divergence between it and the prior:

$$D_{KL}(q_z(\boldsymbol{z}_t|\boldsymbol{x}_{1:t}^0, \boldsymbol{z}_{1:t-1})||p_z(\boldsymbol{z}_t|\boldsymbol{x}_{1:t-1}^0, \boldsymbol{z}_{1:t-1})). \quad (14)$$

Then, putting this together with the aforementioned diffusion objective (equation 7), we have a step-wise dual objective function for our proposed StochDiff:

$$\mathcal{L}_{dual} = \sum_{t=1} D_{KL}(q_z(\boldsymbol{z}_t|\boldsymbol{x}_{1:t}^0, \boldsymbol{z}_{1:t-1})||p_z(\boldsymbol{z}_t|\boldsymbol{x}_{1:t-1}^0, \boldsymbol{z}_{1:t-1})) + \mathbb{E}_{\boldsymbol{x}_t^0, n, \boldsymbol{z}_t}[||\boldsymbol{x}_t^0 - \boldsymbol{x}_\theta(\boldsymbol{x}_t^n, n, \boldsymbol{z}_t)||^2].$$

$$(15)$$

Notice here, we use a data prediction module in the diffusion model rather than the common setting of a noise prediction model. Our rationale is that in previous diffusion models, the added noise in each diffusion step is Gaussian noise with scheduled variances, which makes the noise more predictable. However, the learned prior in our model introduces extra noise into the latent variable, making the noise less predictable. Thus, a data prediction model can be more effective here. We designed an *Attention-Net* that extracts the correlations between dimensions at each time step, and reconstructs the data point with auxiliary prior knowledge. The details are provided in Appendix D.

The model is trained by modelling the time series data in the training set. Once trained, the model is used to autoregressively forecast the future time series with the following steps: (1) model the observations to setup the hidden units, (2) at each future time step obtain the prior knowledge from the hidden units, and (3) sample the data with auxiliary prior knowledge. This sampled data is later used for these computations at the next step. The details of training and forecasting process are provided in Algorithm 1 and 2, and $\mathcal{U}$ denotes the uniform distribution.

### 4.4 IMPROVED POINT-WISE SOLUTION

Similar to other probabilistic models, diffusion models always generate a data distribution instead of point estimation. This introduces the problem of choosing a single result from the distribution for a downstream application. Existing diffusion models directly use the median values as the final results to represent the central tendency of predictions. However, this strategy might introduce errors given a complex data distribution such as mixture of Gaussians. To address this issue, inspired by (Hu et al., 2022), we propose an additional step to train a Gaussian Mixture Model (GMM) on the outputs and identify the largest cluster center as the selected point-wise result.

## 5 EXPERIMENTS

We evaluate our proposed StochDiff on six real-world time series datasets, against several state of the art baseline diffusion models for time series forecasting tasks.

---

**Algorithm 1** Training via Time Series Modelling

---

1: **Input** Training time series data $x_{1:t_0}$.
2: Initialize $h_0 = 0, \mathcal{L}_{total} = 0$.
3: **repeat**
4:     **for** $t = 1$ to $t_0$ **do**
5:         Sample $n \sim \mathcal{U}(\{1, 2, \ldots, N\})$.
6:         Sample $\epsilon \sim \mathcal{N}(0, \boldsymbol{I})$.
7:         Observe $\boldsymbol{x}_t$ as $\boldsymbol{x}_t^0$.
8:         Obtain $p_z, q_z$, and $\boldsymbol{z}_t \sim q_z$ from equation 9, 10.
9:         Obtain reconstructed $\boldsymbol{x}_t^0$ based on $\boldsymbol{z}_t$ and equation equation 13.
10:         Update $h_t$ via $\boldsymbol{f}_\theta(\boldsymbol{x}_t, \boldsymbol{z}_t, h_{t-1})$.
11:         Calculate loss function $\mathcal{L}_{dual}$ in equation 15.
12:         $\mathcal{L}_{total} + = \mathcal{L}_{dual}$.
13:     **end for**
14:     Take gradient descent step on $\nabla \mathcal{L}_{total}$, and update model parameters.
15: **until** converged

---

**Algorithm 2** Autoregressive Forecasting

---

**Require:** trained denoising network $\boldsymbol{x}_\theta$, recurrent network $\boldsymbol{f}_\theta$.
1: **Input** Test time series data $\boldsymbol{x}_{1:T}$.
2: Initialize $h_0 = 0$.
3: **for** $t = 1$ to $t_0$ **do**
4:     Observe $\boldsymbol{x}_t$ as $\boldsymbol{x}_t^0$.
5:     Obtain $\boldsymbol{z}_t \sim q_z$ from equation 10.
6:     Update $h_t$ via $\boldsymbol{f}_\theta(\boldsymbol{x}_t, \boldsymbol{z}_t, h_{t-1})$.
7: **end for**
8: Obtain $h_{t_0}$ (end point of the previous **for** loop).
9: **for** $t = t_0 + 1$ to $T$ **do**
10:     Obtain $\boldsymbol{z}_t \sim p_z$ from equation 9
11:     **for** $n = N$ to $1$ **do**
12:         Sample $\hat{\boldsymbol{x}}_t^{n-1}$ using equation 13.
13:     **end for**
14:     Update $h_t$ via $\boldsymbol{f}_\theta(\hat{\boldsymbol{x}}_t^0, \boldsymbol{z}_t, h_{t-1})$.
15:     Obtain $\hat{\boldsymbol{x}}_t^0$ for $\hat{\boldsymbol{x}}_{t_0+1:T}$.
16: **end for**
17: **return** $\hat{\boldsymbol{x}}_{t_0+1:T}$

---

## 5.1 DATASETS

The time series datasets we used include 4 commonly used datasets in time series forecasting. They mainly contain homogeneous time series that have been collected over a long period, and exhibit some consistence on the temporal dependencies. These data include: *Exchange, Weather, Electricity*, and *Solar*. These datasets containing a long-period time series, we divide each of them into training and testing parts based on their recording time (70% training and 30% testing).Then, we use a sliding window to sample sub-series from each part, and use them as the inputs to the models. More information of these datasets and the sampling details are provided in Appendix C.

In addition to these homogeneous time series data, we also include two heterogeneous time series data from clinical patient datasets to evaluate our model's ability for learning highly stochastic multivariate time series data. They include: (1) *ECochG* (Wijewickrema et al., 2022), which contains 78 patients' records from cochlear implant surgeries and (2) *MMG*[1], which contains uterine magnetomyographic (MMG) signals of 25 subjects. These patient datasets contain high inter and intra variances across different patients, and bring extra challenges to the forecasting task. For these patient datasets, we randomly select 70% patients for training, and use the rest of the patients' data for testing.

---

[1]https://physionet.org/content/mmgdb/1.0.0/

## 5.2 BASELINE METHODS & EVALUATION METRICS

Focusing on the diffusion based models, we compare our proposed StochDiff with existing state-of-the-art (SOTA) diffusion models developed for time series forecasting tasks, including the well-known time series diffusion models: TimeGrad (Rasul et al., 2021), SSSD (Alcaraz and Strodthoff, 2023), and TimeDiff (Shen and Kwok, 2023), as well as the most recent models: TMDM (Li et al., 2024) and MG-TSD (Fan et al., 2024). Details of our model's setup can be found in Appendix D.

To quantitatively evaluate model performance, we use two common metrics, namely Normalized Root Mean Squared Error ($NRMSE$) and Continuous Ranked Probability Score ($CRPS$). The former evaluates the quality of the final point-wise results, and the latter measures the compatibility of the model's predictions with the real data. Here, instead of using original $CRPS$, following (Rasul et al., 2021), we use $CRPS_{sum}$. The details of these metrics can be found in Appendix E. For both metrics, lower values are better.

## 5.3 EXPERIMENTAL RESULTS

Quantitative results are provided in Tables 1 and 2, The best results are marked with bold font. We

Table 1: $NRMSE$ results. Lower values are better.

| Model | Exchange | Weather | Electricity | Solar | ECochG | MMG |
|---|---|---|---|---|---|---|
| TimeGrad | 0.066±0.007 | 0.691±0.017 | 0.043±0.003 | 0.973±0.056 | 1.652±0.025 | 2.977±0.101 |
| SSSD | 0.065±0.006 | 0.755±0.041 | 0.094±0.005 | 1.058±0.053 | 0.965±0.041 | 2.184±0.105 |
| TimeDiff | 0.074±0.002 | 0.702±0.025 | 0.045±0.003 | 0.882±0.044 | 0.916±0.033 | 1.565±0.091 |
| TMDM | 0.063±0.005 | 0.564±0.019 | 0.063±0.004 | 0.829±0.038 | 1.074±0.052 | 5.366±0.114 |
| MG-TSD | 0.075±0.005 | 0.696±0.023 | **0.032±0.002** | **0.756±0.051** | 0.994±0.045 | 1.716±0.087 |
| StochDiff (ours) | **0.052±0.007** | **0.491±0.014** | 0.048±0.002 | 0.812±0.042 | **0.859±0.031** | **1.529±0.094** |

Table 2: $CRPS_{sum}$ results. Lower values are better.

| Model | Exchange | Weather | Electricity | Solar | ECochG | MMG |
|---|---|---|---|---|---|---|
| TimeGrad | 0.010±0.002 | 0.527±0.015 | 0.024±0.002 | 0.416±0.036 | 1.021±0.002 | 1.057±0.041 |
| SSSD | 0.010±0.001 | 0.701±0.047 | 0.065±0.005 | 0.506±0.053 | 0.781±0.003 | 1.372±0.025 |
| TimeDiff | 0.012±0.002 | 0.635±0.019 | 0.036±0.004 | 0.423±0.071 | 0.516±0.002 | 0.948±0.020 |
| TMDM | 0.010±0.001 | 0.618±0.029 | 0.042±0.005 | 0.382±0.039 | 0.703±0.004 | 1.495±0.043 |
| MG-TSD | 0.009±0.000 | 0.596±0.027 | **0.019±0.001** | **0.375±0.067** | 0.683±0.002 | 0.961±0.031 |
| StochDiff (ours) | **0.008±0.001** | **0.521±0.012** | 0.029±0.003 | 0.391±0.052 | **0.435±0.003** | **0.818±0.023** |

can see our proposed StochDiff achieves competitive performances on homogeneous time series data comparing to the existing SOTA methods. However, the existing approaches become ineffective for clinical datasets where individual variability must be taken care of. We highlight the performance of our StochDiff on clinical datasets where the model achieves best results on both datasets and improve the predictive accuracy by around 15% and 14% with respect to the $CRPS_{sum}$. To further assist the assessment, we provide some visual forecasting results on the *ECochG* dataset in Figure 2. As

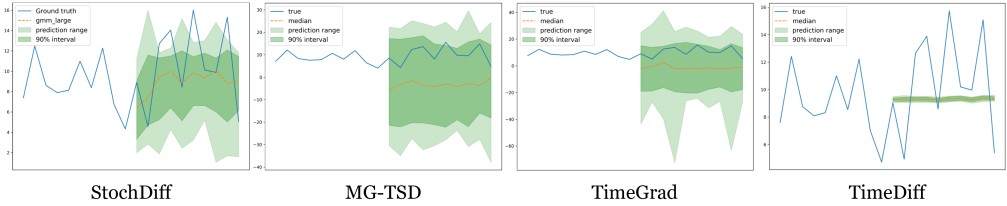

Figure 2: Forecasting results of 4 the best models (based on quantitative results). We display the entire prediction range together with the 90% prediction interval via green bars. The orange dashed lines represent the point-wise results which are medians for baseline methods and the largest centers of fitted GMM for our model.

shown, the baselines fail to capture the variance of the data and tend to predict stationary values for this dataset, and their prediction distribution is either too wide or too narrow. On the other hand, our StochDiff model can learn the dynamics from this highly complex and stochastic data, and is able to narrow its predictions around the true values. Note here, the TimeDiff model was originally designed for point-wise prediction, and thus, its prediction distribution is relatively concentrated. We provide more visual results in Appendix F.

## 5.4 ABLATION STUDY

To investigate the contribution of the different components in our proposed model, we evaluate several basic component models on two clinical datasets: (1) A basic RNN model, *LSTM* and (2) a variational derivation of LSTM, *vLSTM*-$\mathcal{N}(0,1)$ that models the data at each time step with a Standard VAE (Kingma and Welling, 2014), (3) The vLSTM model with diffusion module replacing VAE, *vLSTM-diffusion*, and (4) our proposed *StochDiff* that learns data-driven prior knowledge at each time step. By comparing their performance, we can justify the improvements brought by the step-wise modelling (1,2), the generative diffusion process (2,3) and learned prior knowledge (3,4) for time series forecasting. Details are provided in Table 3. Here, $CRPS_{sum}$ is not applicable for

Table 3: Ablation Study on *ECochG* and *MMG* datasets

| Model | ECochG | | MMG | |
|---|---|---|---|---|
| | NRMSE | CRPS | NRMSE | CRPS |
| LSTM | 1.003±0.042 | - | 3.657±0.138 | - |
| vLSTM-$\mathcal{N}(0,1)$ | 0.982±0.036 | 0.623±0.014 | 3.315±0.114 | 1.767±0.041 |
| vLSTM-diffusion | 0.956±0.035 | 0.505±0.017 | 3.001±0.112 | 1.639±0.032 |
| StochDiff (ours) | **0.859±0.031** | **0.435±0.003** | **1.529±0.094** | **0.818±0.023** |

deterministic *LSTM*. From the results, we can see each component contributes to the performance improvement. The most critical component, prior learning, adds $33,024$ learnable parameters to the original $1065190$ parameters, representing a $3.1\%$ increase. This is negligible in terms of the model's overall capacity. Thus, the performance improvement observed with StochDiff can be attributed primarily due to the inclusion of the step-wise prior knowledge learning procedure, rather than parameter increase. This procedure enhances the model's ability to learn stochastic time series data.

## 5.5 CASE STUDY ON COCHLEAR IMPLANT SURGERY

Real-world application has become a crucial benchmark to evaluate the practical benefits of machine learning models. While many models achieve state-of-the-art results on various datasets, their applicability in real-world scenarios often remains an open question. In this section, we aim at bridging this gap for our model by simulating the Cochlear Implant Surgery process and evaluating the model's ability to assist with the operation.

The ECochG dataset comprises inner ear responses during Cochlear Implant Surgery for each patient. The raw data, containing 218 channels, can be converted into a univariate signal called the 'Cochlear Microphonic' (CM). **Drops** larger than **30%** in the amplitude of the CM signal has been proven to be capable of reflecting the damage to a patient's inner ear structure (Campbell et al., 2016; Dalbert et al., 2016). Thus, researchers have developed models to monitor the CM signal and automatically detect these 'traumatic drops' (Wijewickrema et al., 2022).

Now, with forecasting models, we can attempt to predict these drops even before they occur. To simulate the data stream in real-world surgeries, we use a sliding window to sample the patient's data so that the model can only observe the data up to the current time step. Then, we let the model predict the future steps based on the historical observations. Additionally, considering that in real-world scenarios, the models are expected to work with new data, we use new patients data (out of the aforementioned 78 patients) to test the models' robustness. Figure 3 shows several cases where possible 'traumatic drops' occurred.

Here, we follow the same steps as (Campbell et al., 2016; Dalbert et al., 2016) to convert the predicted raw values into CM amplitudes, and compare with the real CM amplitudes. With these simulations, we can obtain more straightforward insights on the goodness of the models. While the baseline

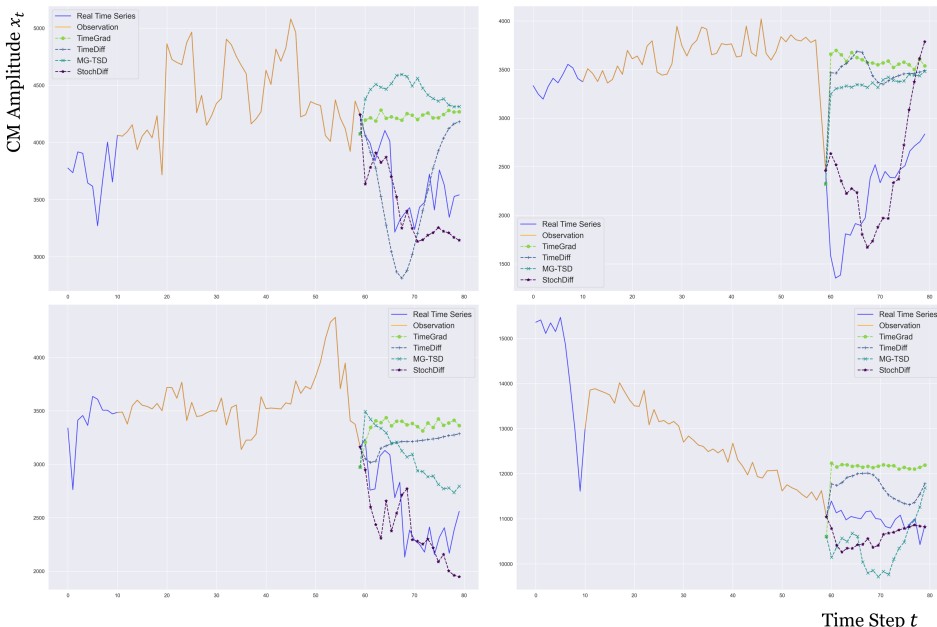

Figure 3: Cochlear Implant Forecasting Simulations. The blue line is the real CM amplitude, and the orange part is the observation that was fed to the model. Forecasting from different models are marked with differently according to the legend inside the plots.

models show some capability to predict the 'traumatic drops', our StochDiff model can forecast these drops in all four cases. This result demonstrates our model's robustness on new data and showcases its potential in a clinical context, for use in Cochlear Implant Surgeries.

To demonstrate the practicability of the models in a real-world surgery scenario, we compare the sampling time of each model in Table 4. Here, we record the time required for each model to generate

Table 4: Sampling Time in seconds. Lower values are better.

| Models | Sampling Time (s) |
|---|---|
| TimeGrad | 2.752±0.121 |
| SSSD | 42.175±0.637 |
| TimeDiff | 38.082±0.312 |
| TMDM | 31.635±0.472 |
| MG-TSD | 3.405±0.139 |
| StochDiff (ours) | **0.9813±0.115** |

100 samples of 5-step predictions for a single time series. We perform 10 independent runs to get the averages and standard deviations. Following common practice, we accelerate the sampling in StochDiff via parallelism and DDIM sampling strategy. From the Table, we can see the StochDiff costs the lowest time among all models. The average sampling time of $0.9813$ second for a 5-step prediction makes it practical to apply StochDiff for a real-time surgery.

## 6 CONCLUSION

In this paper, we propose StochDiff, a novel diffusion based forecasting model for stochastic time series data. By applying step-wise modelling and data-driven prior knowledge, StochDiff is better able to model the complex temporal dynamics and uncertainty from highly stochastic time series data. Experimental results on six real-world datasets demonstrate the competitive performance of StochDiff when compared with state-of-the-art diffusion based time series forecasting models. Additionally, a case study on cochlear implant surgery showcase the model's effectiveness and efficiency in predicting highly stochastic patient data, highlighting its potential in a real-world medical application.

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

## A    MATH FORMULATION

During the reverse diffusion process, when we introduce an additional latent variable into the prior distribution, we have a prior distribution conditioned on $z$, and the equation 1 becomes:

$$p(\boldsymbol{x}^{0:N}) = p(\boldsymbol{x}^N|\boldsymbol{z}) \prod_{n=1}^{N} p_\theta(\boldsymbol{x}^{n-1}|\boldsymbol{x}^n, \boldsymbol{z})p(\boldsymbol{z}). \tag{16}$$

By integrating out all other variables $\boldsymbol{x}^{1:N}$, we can obtain the data distribution $p(\boldsymbol{x}^0)$:

$$p(\boldsymbol{x}^0) = \int_{\boldsymbol{x}^{1:N}} \int_{\boldsymbol{z}} p(\boldsymbol{x}^N|\boldsymbol{z}) \prod_{n=1}^{N} p(\boldsymbol{x}^{n-1}|\boldsymbol{x}^n, \boldsymbol{z})p(\boldsymbol{z})d\boldsymbol{z}d\boldsymbol{x}^{1:N}. \tag{17}$$

The equation 17 defines a latent variable reverse diffusion process, which we call *variational diffusion*. By introducing the time series indexes and prior distribution (equation 9) into the *variational diffusion*, we have the RHS of equation 12:

$$\int_{\boldsymbol{x}_{t_0:T}^{1:N}} \int_{\boldsymbol{z}_{t_0:T}} \prod_{t=t_0}^{T} p(\boldsymbol{x}_t^N|\boldsymbol{z}_t) \prod_{n=1}^{N} p_\theta(\boldsymbol{x}_t^{n-1}|\boldsymbol{x}_t^n, \boldsymbol{z}_t)p_z(\boldsymbol{z}_t|\boldsymbol{x}_{1:t-1}^0, \boldsymbol{z}_{1:t-1})d\boldsymbol{z}_{t_0:T}d\boldsymbol{x}_{t_0:T}^{1:N}. \tag{18}$$

Following the derivation below, we proved equation 18 is the valid formulation of the generative process of *variational diffusion* on *time series forecasting* task, and we name it Stochastic Diffusion Model.

$$\int_{x_{t_0:T}^{1:N}} \int_{z_{t_0:T}} \prod_{t=t_0}^{T} \left( p(x_t^N|z_t) \prod_{n=1}^{N} p(x_t^{n-1}|x_t^n, z_t) \right) p(z_t|x_{1:t-1}^0, z_{1:t-1})dz_{t_0:T}dx_{t_0:T}^{1:N}$$

$$= \int_{x_{t_0:T}^{1:N}} \left( \int_{z_{t_0:T}} \prod_{t=t_0}^{T} p(x_t^{0:N}|z_t)p(z_t|x_{1:t-1}^0, z_{1:t-1})dz_{t_0:T} \right) dx_{t_0:T}^{1:N}$$

$$= \int_{x_{t_0:T}^{1:N}} \left( \int_{z_{t_0:T}} p(x_{t_0:T}^{0:N}, z_{t_0:T}|x_{1:t_0-1}^0, z_{1:t_0-1})dz_{t_0:T} \right) dx_{t_0:T}^{1:N}$$

$$= \int_{x_{t_0:T}^{1:N}} p(x_{t_0:T}^{0:N}|x_{1:t_0-1}^0, z_{1:t_0-1})dx_{t_0:T}^{1:N}$$

$$= p(x_{t_0:T}^0|x_{1:t_0-1}^0, z_{1:t_0-1})$$

## B    GRAPHIC DECOMPOSITION

Figure 4 provides a graphic decomposition of the modelling operations in Figure 1.

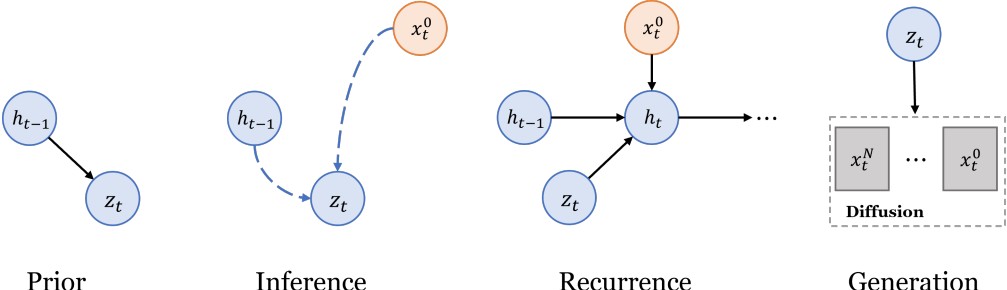

| Prior | Inference | Recurrence | Generation |

Figure 4: Graphic illustration of the modelling operations of StochDiff. (1) Obtaining conditional prior via equation 9. (2) Inference of approximate posterior via equation 10. (3) Hidden state update via sequential model. (4) Data generation via diffusion model.

## C   DATASETS DETAILS

Descriptions for long-period time series datasets are provided below:

- *Exchange*[2], which records the daily exchange rate of 22 countries (to US dollars) from 2000 to 2019.
- *Weather*[3], which consists of records of 21 meteorological reads at 10-minute intervals from 2022 to 2023.
- *Electricity*[4], which records electricity consumption of 321 clients in every 15 minutes from 2012 to 2014. The data in the repository has been converted to an hourly frequency.
- *Solar*[5], which contains solar power production records in year of 2006, data are collected from 56 sensors at Texas State with 5-minute intervals.

To simulate a real-time forecasting scenario, after dividing each dataset with the train-test split, we sample the data using a sliding window with varying size. The model are trained to model the sampled time series data in the training set. Then, during the test time, the model first update the hidden units based on the observations (inside windows) and forecast the future data with specified duration. The statistical details are listed in Table 5.

Table 5: Statistical details of the datasets. Values in parenthesis represents the number of sampled data points.

| Datasets | Window Size | Forecasting | Dimension |
|---|---|---|---|
| Exchange | 3 months (90) | 1 week (7) | 22 |
| Weather | 1 day (144) | 1 day (144) | 21 |
| Electricity | 7 days (168) | 1 day (24) | 321 |
| Solar | 4 days (96) | 1/2 day (12) | 56 |
| ECochG | 1/2 minute (50) | 7 seconds (10) | 218 |
| MMG | 3 seconds (100) | 0.3 second (10) | 148 |

## D   NETWORK DETAILS & EXPERIMENT SETUP

The proposed StochDiff consists of several components that responsible for various learning tasks. The Recurrent backbone (*LSTM*) is the basis of time series modelling: it takes time series data $x_t$ at time $t$ and its corresponding prior knowledge $z_{t-1}$ as inputs, then extracts the temporal dynamics and stores them inside 128-length hidden units $h_t$. A Fully Connected Network (FCN) with non-linear activation function projects $h_{t-1}$ into distributional parameters $\mu, \delta$. Then, another FCN projects these parameters into a 128-length vector $z_t$, representing the prior latent variable. Meanwhile, the approximate posterior is estimated with similar networks that takes both $x_t$ and $h_{t-1}$ as the inputs. The prior knowledge is learnt based on the principle of variational inference and its implementation surrogate, VAE. The diffusion module is responsible for the data reconstruction and generation. The forward diffusion process is mainly involved with mathematical calculations. And during the reverse diffusion process, a data prediction network *Attention-Net* is designed for multivariate time series modelling. The *Attention-Net* (as showed in Figure 5) consists of a series of residual blocks adapted from existing works (van den Oord et al., 2016; Kong et al., 2021; Alcaraz and Strodthoff, 2023). Each residual block contains two different attention mechanisms to generate original data: Firstly, a self-attention is applied to learn channel correlations for $x_t$. Then, a cross-attention properly combines the outputs of self-attention with the conditional latent $z_t$ to integrate the temporal dynamics, uncertainties and multi-modalities into the data prediction process. Finally, a feed forward network projects the outputs of residual blocks into the expected outputs - the predicted data. Through the backpropagation, the model will optimize the parameters in each part to learn the expected components.

---

[2]https://www.kaggle.com/datasets/brunotly/foreign-exchange-rates-per-dollar-20002019/data

[3]https://www.bgc-jena.mpg.de/wetter/

[4]https://github.com/laiguokun/multivariate-time-series-data/tree/master/electricity

[5]https://www.nrel.gov/grid/solar-power-data.html

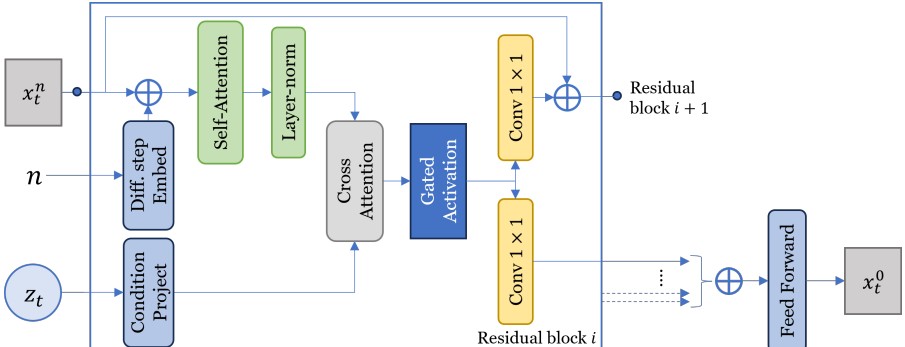

Figure 5: Attention-Net

The model is trained using the Algorithm 1 provided in main text. Noted here, to enhance the optimization process, we reduce the learning rate by $50\%$ whenever loss stops dropping for 10 epochs (patient time). All experiments are undertaken using a High Performance Computer (HPC) with Intel Xeon Gold 6326 CPU (2.90GHz) and NVIDIA A100 GPU (80G).

## E    EVALUATION METRICS

The $NRMSE$ firstly calculates the mean squared error which is the squared differences between estimated future values and the real values. Then it takes the square root of the results and normalize it using the mean of real data to facilitates the different scales of different datasets:

$$NRMSE := \frac{\sqrt{\mathbb{E}((\boldsymbol{x} - \hat{\boldsymbol{x}}))}}{\mathbb{E}(\boldsymbol{x})}.$$

The $CRPS$ measures the quality of the predicted distribution by comparing the Cumulative Distribution Function (CDF) $F$ of the predictions against the real data $x \in \mathbb{R}$:

$$CRPS(F, x) := \int_{\mathbb{R}} (F(y) - \mathbb{I}(x \leq y))^2 dy$$

where $\mathbb{I}(x \leq y)$ is an indicator function, $\mathbb{I} = 1$ when $x \leq y$, and $\mathbb{I} = 0$ otherwise. Following (Salinas et al., 2019), we calculate the $CRPS_{sum}$ for multivariate time series data by summing sampled and real data across dimensions and then averaging over the prediction horizon:

$$CRPS_{sum} = \mathbb{E}_t \left[ CRPS(\hat{F}_{sum}(y), \sum_{i=1}^{d} x_t^{(i)}) \right]$$

where $\hat{F}_{sum}(y) = \frac{1}{S} \sum_{s=1}^{S} \mathbb{I}(x_s \leq y)$ is the empirical CDF with $S$ samples used to approximate the predictive CDF. This score has be proved to be a proper score for multivariate time series data (de Bézenac et al., 2020).

## F    VISUAL RESULTS

Figure 6 provides more visual results on ECochG dataset from *StochDiff*, *MG-TSD*, and *TimeGrad*. Here each row showcases the forecasting results of the same data sample from three models. The predictions of *StochDiff* is closely distributed around the real data while the other models produce either deviated or highly uncertain results.

## G    LIMITATIONS & FUTURE PERSPECTIVES

Several directions can be investigated in the future to further improve the current work:

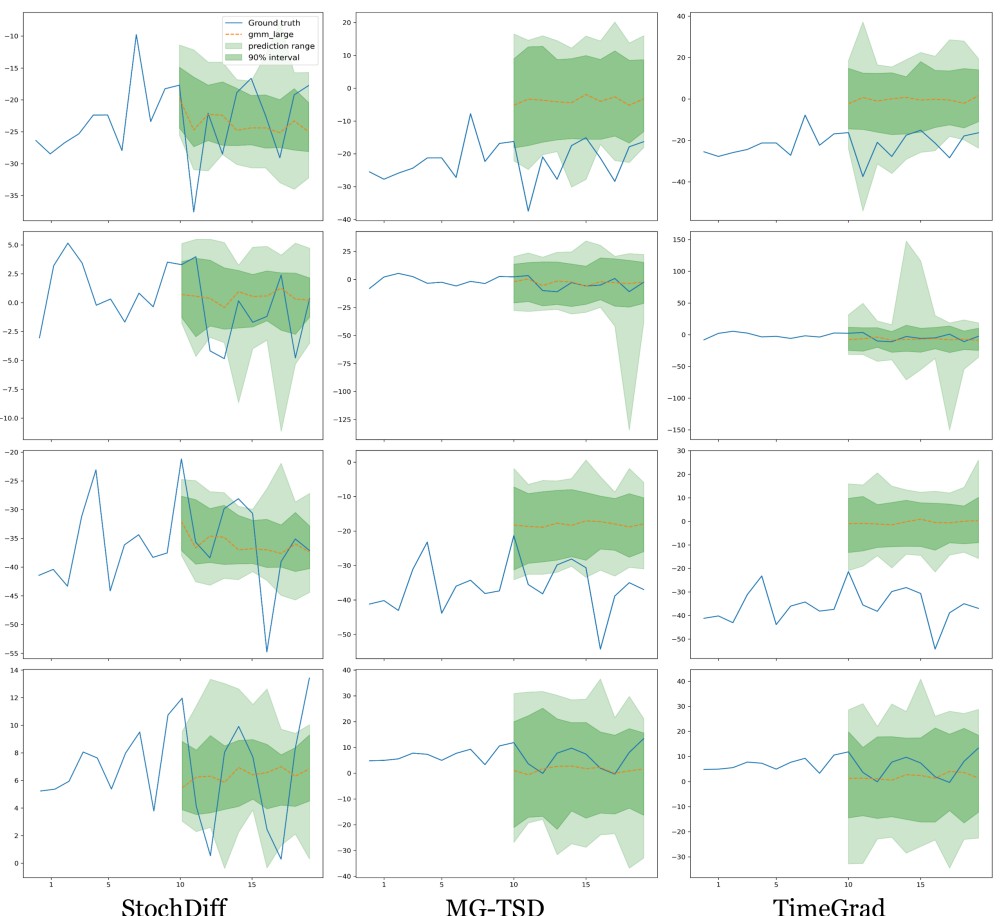

Figure 6: Visual Results on ECochG dataset.

- **Prior Knowledge Learning:** The current prior knowledge at each time step is provided to diffusion model as an auxiliary condition during the generation process. This design can be improved by directly integrating this prior knowledge into latent variable and using it as the end/beginning point of diffusion process (gil Lee et al., 2022). On the other hand, the current model projects the learnt hidden units into corresponding prior knowledge using a MLP with non-linear activation function. A more robust architecture may possibly improve the prior knowledge learning comparing to this design.

- **Sequential Backbone:** The current model uses LSTM as the sequential backbone, which may lead to one limitation: Although the local variance/uncertainty at each step can be efficiently extracted, the long term features still rely on the LSTM to capture. Thus, as we explained in section 5.3, on long-period homogeneous datasets (e.g. Electricity), our model did not show superior performance comparing with baselines, such as MG-TSD or TMDM, which focus on these homogeneous datasets to develop certain mechanisms to extract temporal features (e.g. MG-TSD proposed to learn temporal features with multiple granularity). In the future, a combination of different sequential modules can be proposed to learn different features in time series data (e.g., Liu et al. (2024) proposed a network consists of TCN and Transformer to simultaneously learn global and local features).

- **Efficiency:** The current model apply parallelism and DDIM (Song et al., 2021) sampling strategy during the forecasting stage to enhance the sample speed. This strategy improves the time efficiency at the cost of increased memory requirements. In the future, a lightweight model design could be explored to achieve efficiency in both time and memory.

- **Real-world Implication:** As the current study is for research purpose, the model was only tested on limited datasets. More comprehensive testing and validation are required to prepare it for use in real-world surgeries. It is also important to follow ethical practices when using generative models to avoid any negative human impact. E.g. closely integrating the model with human in the loop oversight.

