# OpenReview forum: "Stochastic Diffusion: A Diffusion Based Model for Stochastic Time Series Forecasting"
_ICLR.cc/2025/Conference — Submitted to ICLR 2025_

### Official Review · Reviewer_iATK · 2024-10-28

**Soundness:** 1
**Presentation:** 2
**Contribution:** 2
**Rating:** 3
**Confidence:** 4

**Summary:**

This paper introduces the StochDiff model, an approach for generative time series forecasting that leverages diffusion processes and stochastic latent spaces to capture the high variability of stochastic time series data. Unlike previous methods that rely on sequential models and unimodal latent variables, StochDiff integrates a diffusion module at each time step and learns a data-driven prior, effectively modeling complex temporal dynamics and multi-modal characteristics.

**Strengths:**

The introduction and background sections of the paper are well-written and easy to follow, providing a clear overview of the main ideas and approach.

**Weaknesses:**

The methodology part of the paper is quite confusing and even contains several errors. Please refer to the "Questions" section for details.

**Questions:**

1. Why is a Fully Connected Network (FCN) necessary in the Prior Encoder to project the sampled random variable into $ z_t $? Wouldn't it be possible to directly match the dimensions of $\hat{\mu}$ and $\hat{\sigma}$ to $z_t$? What is the rationale behind this approach?

2. "With equation (9), the prior distribution at... which enables it to (1) encode more informative representations, (2) better approximate the real data distributions, and subsequently, (3) benefit the time series data modeling." This statement lacks any actual explanation. Why does using the random variable $z$ as guidance confer these advantages? Can you provide any explanation?

3. Why is equation (10) necessary? If I understand correctly, equation (10) is ultimately used in the dual objective function. However, according to equation (9), $z_t$ should represent information from times $1:t-1$ and should not include information from $x_t$. Therefore, the posterior distribution of $z_t$ given $x_t$ expressed in equation (10) should not be used.

4. Equation (11) also has significant issues. In equation (11), the mean and variance depend on $h_{t-1}$; however, according to equation (9), $h_{t-1}$ should only depend on $z_{1:t-1}$ (as per the recursive architecture) and not on $z_t$. Hence, the correct expression for equation (11) should be $p_\theta(x_t^{n-1}|x_t^n, z_{1:t-1})$ rather than one that depends on $z_t$. Similarly, the formulations $p_\theta(x_t^{n-1}|x_t^n, z_{t})$ in equation (12) and $x_\theta(x_t^n, n, z_t)$ in equation (13) are incorrect.

5. "The prior knowledge learning is associated with the variational inference that optimizes the approximate posterior $q_z(z_t|x^0_{1:t}, z_{1:t−1})$ by minimizing the KL divergence between it and the prior." This statement is also quite confusing. First, why optimize the approximate posterior? If I understand correctly, the proposed method does not use the approximate posterior; it only uses $p_z(z_t|x^0_{1:t-1}, z_{1:t-1})$. Second, according to the principles of variational inference, we should aim to minimize the KL divergence between the approximate posterior and the true posterior, not the prior.

6. Efficiency concerns. The proposed method incorporates the diffusion process at each time step, raising a natural question about its efficiency. Given that each time-step observation involves multiple steps of the diffusion process, and with many time steps requiring the same operation, wouldn't this approach be highly inefficient?

---

> ### Author Response · Authors · 2024-11-19
> **Reply to Reviewer iATK (1st part)**
>
> We would like to thank the reviewer for spending time and providing these comments. We are willing to provide more explanation for our method.
>
> - **Why is a Fully Connected Network (FCN) necessary in the Prior Encoder to project the sampled random variable $z_t$? Wouldn't it be possible to directly match the dimensions of parameters to latent variable? What is the rationale behind this approach?**
>
> Answer: The FCN with a non-linear activation function used in our model functions as a projector during the entire prior learning process. This design ensure flexibility and expressiveness of the model as it can learn non-linear relationships between the variables, leading to a richer latent space. Secondly, it is more effective to use a network for high-dimensional and multi-modal data comparing to matching the dimensions of variables. Finally, as a part of the deep learning model, embedding with FCN allowing end-to-end training and optimization. Thus, although it is true that we can directly match the dimension of the parameters to the latent variables, a learning approach should be a better option here. This is also a common design in existing works [1,2,3].
>
> - **"With equation (9),the prior distribution at... which enables it to (1)encode more informative representations,(2)better approximate the real data distributions, and subsequently,(3) benefit the time series data modeling." This statement lacks any actual explanation. Why does using the random variable as guidance confer these advantages?**
>
> Answer: We are willing to provide more details on the advantage of a dynamic prior distribution in our approach:
>
> (1) Comparing to the common fixed prior (e.g. $\mathcal{N}(0,1)$), a step-wise dynamic prior allows the model to adapt its understanding of latent variables based on the historical data up to that point. This adaptability enables the latent variables to encapsulate relevant temporal dynamics and correlations specific to the period being modeled, thus **encoding richer and more informative representations** of the data.
>
>  (2) Most real-world time series data often exhibit non-stationary behaviors (changes in mean, variance, seasonality), a dynamic prior enables the model to adjust continually, maintaining **alignment with the actual data distribution**.
>
>  (3) Previous two points together contribute to a better time series modelling: As the model can quickly integrates fresh insights based on the dynamic, data-driven latent variable. The continual updating mechanism also helps in maintaining a high level of accuracy in data modelling, as the parameters are always optimized based on the latest available data.
>
>  Using random variables $z_t$ during time series modelling transforms the traditional approaches from merely analyzing temporal patterns to understanding and leveraging the underlying stochastic processes that generate the data. With the dynamic, data-driven priors, the model is responsive and adaptive to the most recent observation, leading to a more accurate data modelling process.
>
> References:
>
> [1] D. P. Kingma and M. Welling. Auto-encoding variational bayes. 2014.
>
> [2] J. Chung, et al. A recurrent latent variable model for sequential data. 2015.
>
> [3] A. Desai, et al. TimeVAE: A Variational Auto-Encoder for Multivariate Time Series Generation. 2021.

---

> > ### Author Response · Authors · 2024-11-19
> > **Reply to Reviewer iATK (2nd part)**
> >
> > - **Why is equation (10) necessary? If I understand correctly, equation (10) is ultimately used in the dual objective function.However, according to equation (9), $z_t$ should represent information from $1:t-1$ times and should not include information from $x_t$. Therefore, the posterior distribution of $z_t$ given $x_t$ expressed in equation (10) should not be used.**
> >
> > - **First,why optimize the approximate posterior?If I understand correctly,the proposed method does not use the approximate posterior; it only uses prior $p_z$. Second, according to the principles of variational inference,we should aim to minimize the KL divergence between the approximate posterior and the true posterior, not the prior.**
> >
> > Answer: We noticed, from these two questions, there might be some confusions and misunderstandings related to the variational inference and its famous application, variational autoencoder (VAE) [1]. We are willing to provide more details about the VAE to help with the understanding of our approach.
> >
> > First of all, we would like to clarify the roles of equation (9) and (10). The equation (9) represents the prior distribution of the latent variable $z_t$ at time step $t$:  $p_z(z_t|x_{1:t-1}, z_{1:t-1})$. This prior distribution encapsulates the model's beliefs about the latent state before observing the current data point $x_t$. On the other hand, the equation (10) is the approximate posterior of $z_t$ given the current observation and the historical data $q_z(z_t|x_{1:t}, z_{1:t-1})$. It reflects a revision of the belief about the latent state after the current observation $x_t$ included.
> >
> > The **necessity of equation (10)** arises from the fundamental aim of variational inference to approximate the true posterior as closely as possible for better model learning and inference. Due to the intractability of true posterior, a simpler posterior from a known distribution family (e.g. Gaussian) is use to approximate the true one. By updating the latent variable's distribution to include information from $x_t$, equation (10) provides a more accurate and updated representation of the latent state based on the full context of observed data up to that point. So, optimizing the approximate posterior in variational inference models ensures the accurate estimation of the intractable true posterior, leading to a better modelling of the time series data.
> >
> > The different **optimization targets** between *variational inference concept* and the *practical implement* come from the proposed surrogate in VAE paper. Again, due to the intractability of true posterior, it is infeasible to directly calculate the divergence between approximation and the true one. Following the famous surrogate proposed by VAE, the evidence lower bound (ELBO) is applied here to indirectly minimize the original objective. This is includes a KL divergence between $q_z$ and $p_z$ and a likelihood of the observed data under the model. The variational inference ensures that maximizing the ELBO is equivalent to minimizing the KL divergence between the approximate posterior and the true posterior. Then, during the sampling (or generation) stage, the model samples only from the prior $p_z$ because, at this stage, the model uses the learned distributions to generate or predict new outcomes based on learned patterns, without additional input data to update these beliefs.
> >
> > We hope this detailed explanation of variational inference and the ELBO method from VAE can address the reviewer's concern related to the prior and approximate posterior of the latent variable $z_t$ and the optimization target. We have also updated the corresponding part in section 4.1 to better state the role of Eq (10) and the theoretical background associated with the prior knowledge learning.

---

> > > ### Author Response · Authors · 2024-11-19
> > > **Reply to Reviewer iATK (3rd part)**
> > >
> > > - **Equation (11) also has significant issues. In equation (11), the mean and variance depend on $h_{t-1}$. however, according to equation (9), $h_{t-1}$ should only depend on $z_{1:t-1}$ (as per the recursive architecture) and not on $z_t$. Hence, the correct expression for equation (11) should be $p_{\theta}(x_t^n|x_t^{n-1}, z_{1:t-1})$ rather than the one use $z_t$.**
> > >
> > > Answer: There might be some misunderstanding on the learning procedure and the dependencies of the proposed model. We are willing to provide more details for a better understanding.
> > >
> > > Let's first go through the critical parts and clarify their dependencies: (i) the hidden units $h_{t-1}$ is learned from the recurrent model (represented by function $f_{\theta}$) and is expressed in equation (9) as $h_{t-1} = f_{\theta} (x_{t-1}^0, z_{t-1}, h_{t-2})$. So $h_{t-1}$ is depended on previous information $x_{t-1}^0, z_{t-1}, h_{t-2}$. (ii) the prior distribution of latent variable $z_t$ is formulated in equation (9) as a Gaussian $\mathcal{N}(\mu_{\theta}(h_{t-1}),\delta_{\theta}(h_{t-1}))$. Here, $z_t$ is depended on $h_{t-1}$, and by extension, depended on information up to $t-1$ time: $x_{1:t-1}^0, z_{1:t-1}$. Thus, equation (9) originally written as $p_z(z_t|x_{1:t-1}, z_{1:t-1})$ can be also written as $p_z(z_t|h_{t-1})$.
> > >
> > > With the detailed explanation of hidden units and latent variable, we can conclude that $z_t$ is sampled from a prior distribution that depends on historical information $h_{t-1}$, and $h_{t-1}$ does not depend on $z_t$, so **no circular dependency occurs**. During the forecasting stage, the proposed model *first samples the latent variable $z_t$* from prior ($p_z(z_t|x_{1:t-1}, z_{1:t-1})$ or $p_z(z_t|h_{t-1})$) and use it as a *condition for the diffusion denoising process* (reverse diffusion), which is *formulated as equation (11): $p_{\theta}(x_t^{n-1}|x_t^n,z_t)$*. The usage of equation (11) is consistent with the model's aim to capture both the accumulated historical information (via $h_{t-1}$) and the predicted stochastic variations at time $t$ (via $z_t$).
> > >
> > > The detailed formulation of equation (11) is developed from the common reverse diffusion in existing work which we provided in equation (3): $p_{\theta}(x^{n-1}|x^n) := \mathcal{N}(x^{n-1}; \mu_{\theta}(x^n, n), \delta_{\theta}(x^n, n))$.
> > >
> > >  We acknowledge that equation (11) in our paper may have caused confusion regarding the dependencies of different variables. Considering that $z_t$ has already encapsulated the historical information through its dependency on $h_{t-1}$, we simplify the equation (11) to eliminate any redundancy and enhance clarity: $p_{\theta}(x_t^{n-1}|x_t^n,z_t) = \mathcal{N}(x_t^{n-1}; \mu_{\mathcal{C}}(x_t^n, n, z_t), \delta_{\mathcal{C}}(x_t^n, n, z_t))$, where $\mu_{\mathcal{C}}, \delta_{\mathcal{C}}$ are functions that compute the parameters of the Gaussian distribution. They integrate $z_t$ into corresponding parameters via function $\mathcal{C}$ (we use cross-attention in our model). Additionally, to help with understanding the operation flow of our model, we added a graphic illustration of the model's operation in Appendix B.
> > >
> > > - **Efficiency concerns.**
> > >
> > > Answer: Although we implement diffusion module at each time step, it does not necessarily mean the modelling time would highly increase. As we store the hidden units $h_t$ at each step, with new observation, we just need to update one step of modelling operation to update hidden information and the latent variable (based on the $h_{t-1}, x_t$). Additionally, we also use parallelism (simultaneously generation $n$ samples) and DDIM (reduce reverse diffusion steps) to speed up the generation process. In section 5.5, we provided a brief demonstration of the time consumptions from different models and showed that our model can generation 100 samples of 5-step prediction within 1 second, highlighting its potential in real-world scenario.

---

> > > > ### Author Response · Authors · 2024-11-25
> > > > **Gratitude for the review**
> > > >
> > > > Once again, we would like to express our sincere appreciation for the reviewer's efforts and the time dedicated to providing comments for our work.
> > > >
> > > > At this stage, we have addressed most of the questions raised by the other reviewers. As we approach the end of the discussion period, we kindly ask the reviewer to let us know if we have addressed the pointed concerning parts? If there are still any unresolved issues or aspects requiring further clarification, we would be happy to provide additional explanations to address them.

---

### Official Review · Reviewer_oUW5 · 2024-11-02

**Soundness:** 3
**Presentation:** 3
**Contribution:** 2
**Rating:** 6
**Confidence:** 3

**Summary:**

This paper develops a diffusion based generative model for capturing time series, using a step-wise, data-driven prior specifically dedicated for stochastic time sequences modeling. In particular, the authors demonstrate the superior performance of the studied method using highly stochastic real-world time sequence data. Moreover, the proposed StochDiff can capture individual variability and make multi-step prediction, which might be unseen properties have not been realized before.

**Strengths:**

- Novelty:
This is the first attempt to combine diffusion processes into capturing time series, using a step-wise, data-driven prior specifically dedicated for highly stochastic time sequences modeling.

- Quality:
Overall, the paper is well written. The main techniques are well documented.

**Weaknesses:**

There might be some novel technical challenges that have not thoroughly discussed for modeling time series data, using diffusion generative models. I would expect to see some further discussion such as the limitations of the work in capturing highly-stochastic time series, or any other types of sequential data. I did not find other weaknesses or flaws.

**Questions:**

Can the authors discuss the limitations of the work in capturing highly-stochastic time series, or any other types of sequential data?

---

> ### Author Response · Authors · 2024-11-19
> **Reply to Reviewer oUW5**
>
> We would like to thank the reviewer for these valuable comments. We now answer the reviewer's question.
>
> - **Can the authors discuss the limitations of the work in capturing highly-stochastic time series, or any other types of sequential data?**
>
> Answer: We provided a brief summary of the possible limitations and future works of the current model in Section F. We pointed out three possible limitations of the current model: (1) prior knowledge provided as condition in generative stage, (2) sequential backbone is a basic LSTM, (3) real-world validation. We are willing to go through these points in detail and provide more possible limitations and corresponding improvements:
>
> (1) In our model, the prior knowledge is provided as a condition during the generative process. Although this is a common design at current stage (applied by most existing works on diffusion model which are cited in paper), further improvement is possible as a previous work investigated the directly usage of a learnt prior as the start point of the reverse diffusion process. This improvement may probably increase the efficiency of the diffusion process. On the other hand, the current model projects the learnt hidden units into corresponding prior knowledge using a MLP with non-linear activation function. A more robust architecture may possibly improve the prior knowledge learning comparing to this design.
>
> (2) Our model uses LSTM as the sequential backbone, which lead to one limitation of the current model: as we designed it to model time series at each step, the local variance/uncertainty can be efficiently extracted. However, the long term features still relies on the LSTM to capture. Thus, as we explained in section 5.3, on long-period homogeneous datasets (e.g. Electricity), our model did not show superior performance comparing with baselines, such as MG-TSD or TMDM, which focus on these homogeneous datasets to develop certain mechanisms to extract temporal features (e.g. MG-TSD proposed to learn temporal features with multiple granularity). In the future, a combination of different sequential modules can be proposed to learn different features in time series data. For example, in Time-Transformer, a network consists of TCN and Transformer is proposed to simultaneously learn global and local features.
>
> (3) As the current study is solely for research purpose, the model was only tested on limited datasets. To apply it for real-world tasks like human surgeries, more comprehensive testing and validations are required to guarantee the patients' safety.
>
> (4) The current model apply parallelism and DDIM sampling strategy during the forecasting stage to enhance the sample speed. This strategy achieve the time efficiency with the cost of increasing memory requirement. In the future, a light-weight model design can be explored to achieve efficiency in both time and memory
>
> We have update the Section F to include more details we provided here.

---

> > ### Comment · Reviewer_oUW5 · 2024-11-22
> > **RE**
> >
> > Ive read your feedback, and want to thank you. I stick with my score.

---

### Official Review · Reviewer_4vTj · 2024-11-02

**Soundness:** 3
**Presentation:** 3
**Contribution:** 2
**Rating:** 6
**Confidence:** 4

**Summary:**

The authors describe a novel model (StochDiff) for probabilistic time series forecasting. It combines a sequence-to-sequence approach consisting of an LSTM network on the input and on the output side.
The hidden LSTM states $h_t$ are also used to train two additional encoders (one taking the past state $h_{t-1}$, one taking the past state $h_{t-1}$ and the current input $x_t$) via two FCNs yielding the latent state $z_t$. Both serve as input to a standard diffusion model which aims to decode the encoded/latent state $z_t$ to $x_t$  (stochastically) by conditioning on this state and are jointly.
The aggregated hidden state at the time of the last observation $h_{t_0}$ is then used to predict $z_{t>t_0}$ utilizing the LSTM. The obtained latent state is decoded by the trained diffusion model. Optionally, the samples produced by the diffusion model are clustered and aggregated to the largest cluster center by a subsequent GMM.
In experiments, the model shows competitive or slightly better performance compared to SOTA models. The authors conclude that the performance is even better in “highly stochastic” (clinical) dataset and complete the paper with a case-study on cochlear implant data.

**Strengths:**

The authors describe a model which contains interesting aspects. The usage of diffusion models in the proposed way is novel, also the architecture with the two encoders. The model is applied to standard datasets and an interesting non-standard case.

**Weaknesses:**

The building blocks of the model, the idea of RNN/sequence-to-sequence processing and the derivation of the properties of the diffusion model are pretty standard.

**Questions:**

p1/2: "integration of the diffusion into the modelling process" vs. "posthoc": unclear, isn't it more about recurrent prediction or some other model?

p4:  The figure does not completely fit to the text and the legend (p5: "blue dash lines", the arrow from $z_t$ to $h_t$ is not clear, the "inference" arrows are not complete, green arrows are not defined etc.)

p5, eq 11: is the complicated form needed?

p6, algorithm 1. There are two $z_t$ symbols as results from the two encoders.

p14, section C: please provide information about the Attention-Net which is crucial for the performance of the diffusion process.

---

> ### Author Response · Authors · 2024-11-19
> **Reply to Reviewer 4vTj**
>
> We would like to thank the reviewer for these valuable comments. We now answer the reviewer's questions point by point.
>
> - **"integration of the diffusion into the modelling process" vs."posthoc": unclear, isn't it more about recurrent prediction or some other model?**
>
> Answer: The main difference between our approach and the existing ones, as we mentioned in introduction, lies in the modelling stage. The existing methods **solely use sequential networks** (e.g. RNN, Transformer) during the modelling stage to learn the temporal features. The diffusion module is **only applied to generate future values** conditioned on the learnt features. Thus, the diffusion in these approaches is a **posthoc** style. However, in our model, the **diffusion is integrate into modelling stage** to help the sequential model extract the uncertainty and learn the multi-modal distribution of complex time series data from the observation. These lead to a better modelling of highly stochastic time series data, enabling the model to produce more accurate prediction of the future values.
>
> We use the Table below to highlight the difference of the "integration of the diffusion into the modelling process" and the "posthoc":
> | Diffusion involved |   Modelling  |  Forecasting |
> |:------------------:|:------------:|:------------:|
> |   Existing models  |   $\times$   | $\checkmark$ |
> |        Ours        | $\checkmark$ | $\checkmark$ |
>
> - **The figure does not completely fit to the text and the legend**
>
> Answer: We have update the figure to fit the text description and the legend.
>
> - **eq 11: is the complicated form needed?**
>
> Answer: Eq 11 formulates the reverse diffusion process of our approach and contrasts it to the one in normal diffusion models (provided in eq 3). However, as suggested by reviewer, we simplify its formulation to eliminate any redundancy and enhance clarity: $p_{\theta}(x_t^{n-1}|x_t^n,z_t) = \mathcal{N}(x_t^{n-1}; \mu_{\mathcal{C}}(x_t^n, n, z_t), \delta_{\mathcal{C}}(x_t^n, n, z_t))$, where $\mu_{\mathcal{C}}, \delta_{\mathcal{C}}$ are functions that compute the parameters of the Gaussian distribution. They integrate $z_t$ into corresponding parameters via function $\mathcal{C}$ (cross attention in our model).
>
> - **algorithm 1. There are two $z_t$ symbols as results from the two encoders.**
>
> Answer: We have update the Algorithm 1 to avoid any confusions. The corresponding step now become "Obtain $p_z, q_z$ and $z_t\sim q_z$ from equation (9), (10)".
>
> - **section C: please provide information about the Attention-Net which is crucial for the performance of the diffusion process.**
>
> Answer: We have updated section C to include more details about Attention-Net: The Attention-Net consists of a series of residual blocks adapted from existing works ([1,2,3], etc.). Each residual block contains two different attention mechanisms to generate original data: Firstly, a self-attention is applied to extract channel correlations for $x_t$. Then, a cross-attention combines the outputs of self-attention with the conditional latent $z_t$ to include the temporal dynamics, uncertainties and multi-modalities into the data prediction process. Finally, a feed forward network projects the outputs of residual blocks into the expected outputs - the predicted data. Through the backpropagation, the model will optimize the parameters in each part to learn the expected components.
>
> In addition to the text description, we also include a figure to visualize the network structure. This can be found in Section C in updated paper.
>
> References:
>
> [1] A. van den Oord, et al. Wavenet: A generative model for raw audio. 2016.
>
> [2] Z. Kong, et al.  Diffwave: A versatile diffusion model for audio synthesis. 2021.
>
> [3] J. L. Alcaraz and N. Strodthoff. Diffusion-based time series imputation and forecasting with structured state space models. 2023.

---

> > ### Author Response · Authors · 2024-11-25
> > **Gratitude for the review**
> >
> > We would like to express our sincere appreciation for the reviewer's efforts and the time dedicated to providing valuable comments for our work.
> >
> > As we approach the end of the discussion period, we would like to know if we have addressed the reviewer's concerns? If there are still any unresolved issues or aspects requiring further clarification, we would be happy to provide additional explanations to address them.

---

> > > ### Comment · Reviewer_4vTj · 2024-11-26
> > > **Reply**
> > >
> > > Thank you for your elaborate reply which answers my questions.

---

### Official Review · Reviewer_aF5F · 2024-11-03

**Soundness:** 2
**Presentation:** 2
**Contribution:** 2
**Rating:** 5
**Confidence:** 3

**Summary:**

The paper presents Stochastic Diffusion (StochDiff), a novel diffusion-based model specifically designed for stochastic time series forecasting. Unlike existing models, StochDiff incorporates the diffusion process directly into each time step during the modeling stage, rather than applying it as a post-hoc operation. This design allows StochDiff to effectively capture complex temporal dependencies and the stochastic nature of high-variability time series. The model employs a data-driven, step-wise prior, with particularly strong performance demonstrated on heterogeneous clinical data.

**Strengths:**

​1. **Innovative Integration of Diffusion in Sequential Modeling**: The authors embed the diffusion process into each time step within the model, enhancing its ability to capture dynamic stochastic behaviors and improving its adaptability to high-variability time series data.

​2. **Data-Driven Prior Knowledge**: The model introduces a data-driven prior, allowing it to learn detailed patterns across multiple time steps and achieve more accurate, contextually relevant predictions.

​3. **Robust Experimental Support**: Extensive testing across various datasets highlights StochDiff’s high performance, especially with clinical data, showing an approximate 15% improvement in predictive accuracy over current state-of-the-art methods.

**Weaknesses:**

​1. **Insufficient Explanation of Fast Sampling Performance**: While the paper claims that StochDiff achieves faster sampling than other models, it lacks sufficient explanation of the underlying mechanisms or optimizations. It’s challenging to understand why the architectural choices enable faster sampling.

​2. **Reliance on Fully Connected Networks (FCN)**: The model relies exclusively on MLPs (FCNs) for encoding, which may raise concerns about robustness and scalability in complex time series settings. The FCN-only ablation does not fully justify the architecture’s solidity; for instance, the observed performance decline when removing parts of the framework might stem from the shallow network’s reduced parameter count rather than validating the effectiveness of the design.

**Questions:**

​1. Why is only an MLP used for prior encoding? Given the complexity of time series data, why does the architecture rely solely on MLPs (FCNs) for prior knowledge encoding? Would a more robust structure enhance the accuracy of the prior?

​2. What specific mechanisms contribute to the claimed fast sampling times? Please clarify if there are specific optimizations or architectural choices that enable StochDiff to achieve faster sampling speeds.

​3. Are the latent variables $z_t$ in the training process derived from Equation (9) or Equation (10)? The design and practical use of Equations (9) and (10) are not clearly explained, making it challenging to understand how and why these formulations are applied.

---

> ### Author Response · Authors · 2024-11-19
> **Reply to Reviewer aF5F**
>
> We would like to thank the reviewer for these valuable comments, we now answer reviewer's questions point by point.
>
> - **Why is only an MLP used for prior encoding? Would a more robust structure enhance the accuracy of the prior?**
>
> Answer: There might be some misunderstanding of our model's architecture, we would like to explain more details for a better understanding. The MLP (together with a non-linear activation function) in our model functions as a projector during the entire prior learning process. Overall, the recurrent model firstly extracts the hidden units $h_{t-1}$ at each time step. Then the MLPs with an activation function are applied to project the $h_{t-1}$ into corresponding parameters $\mu, \delta$ and further project parameters into final variable $z_t$ with demanded dimensions. **Thus, more precisely, the prior knowledge is encoded by both the sequential model and the MLP, rather than solely by MLP**. We notice that our description in Section C may cause this misunderstanding. We have updated the corresponding sentences to avoid any further confusion.
>
> As to the network selection, we acknowledge the reviewer's suggestions on the model design: a more robust structure may further enhance the prior learning procedure. In our work, we are following previous works (VAE, VRNN, TimeVAE) [1,2,3] to use MLPs with activation function to project the hidden units and parameters into final latent variable $z$. We have added this valuable suggestion into our future work (Section F) and will try other model design in the future.
>
> - **What specific mechanisms contribute to the claimed fast sampling times?**
>
> Answer: The content on sampling time is briefly discussed due to the limited space. We are willing to provide more details about this part. The main strategy leads to our fast sampling results is the parallelism, as we design the model to generate $n$ samples simultaneously. On the other hand, we also apply the DDIM [4] method to reduce the reverse diffusion steps, enabling a faster sampling.
>
> - **Are the latent variables in the training process derived from Equation (9) or Equation (10)? The design and practical use of Equations (9) and (10) are not clearly explained, making it challenging to understand how and why these formulations are applied.**
>
> Answer: We notice the confusing part in the mentioned equations and their relations to the latent variable learning. We would like to provide more explanations to make these clear:
>
> First of all, the equation (9) is used to compute the prior distribution of latent variable $z_t$, which is conditional on historical data $x_{1:t-1}$ and the corresponding latent variables $z_{1:t-1}$, $p_z(z_t|x_{1:t-1}, z_{1:t-1})$
>
> In the meantime, equation (10) is used to compute an approximated posterior distribution of $z_t$, which now also conditional on the real data at current step $x_t$, $q_z(z_t|x_{1:t}, z_{1:t-1})$
>
> The idea of variational inference is applied here to minimize the divergence between the approximate posterior $q_z$ and the true posterior $p(z_t|x_{1:t}, z_{1:t-1})$, in order to find the best approximation of the true posterior. However, due to the intractability of the true posterior, it is infeasible to directly calculate the divergence between approximation and the true one. Following the famous surrogate proposed by VAE [1], the evidence lower bound (ELBO) is applied here to indirectly minimize the original objective. This is includes a KL divergence between $q_z$ and $p_z$ and a likelihood of the observed data under the model. The variational inference effectively ensures that maximizing the ELBO is equivalent to minimizing the KL divergence between the approximate posterior and the true posterior.
>
> We have updated several contents that may cause the confusion of latent variable learning, including the section 4.1 and the Algorithm 1.
>
> References:
>
> [1] D. P. Kingma and M. Welling. Auto-encoding variational bayes. 2014.
>
> [2] J. Chung, et al. A recurrent latent variable model for sequential data. 2015.
>
> [3] A. Desai, et al. TimeVAE: A Variational Auto-Encoder for Multivariate Time Series Generation. 2021.
>
> [4] J. Song, et al. Denoising diffusion implicit models. 2021

---

> > ### Comment · Reviewer_aF5F · 2024-11-21
> >
> > Thanks for your response.
> >
> > For question 1, it seems that my concern remains unaddressed. Specifically, my point was that "The FCN-only ablation does not fully justify the architecture’s solidity; for instance, the observed performance decline when removing parts of the framework might stem from the shallow network’s reduced parameter count rather than validating the effectiveness of the design."
> >
> > For question 2, from my perspective, using a batch size larger than 1 and employing DDIM are standard practices in Diffusion models and do not appear to introduce novelty.
> >
> > For question 3, I now have a better understanding of the workflow of your method. However, the explanation in the paper regarding why this approach is necessary still feels insufficient—at least to me.
> >
> > Thanks again for your response. I'll retain my original score.

---

> > > ### Author Response · Authors · 2024-11-22
> > > **Response to further comments**
> > >
> > > We would like to thank the reviewer for the further comments and clarifying the concerning part.
> > >
> > > (1). We acknowledge the concern that "the observed performance decline in the FCN-related ablation studies might stem from a reduction in parameter count rather than the effectiveness of the proposed framework". To clarify, we provide the parameter counts for the two structures evaluated on the ECochG dataset under the same hyperparameter settings as stated in the paper. Specifically: vLSTM-diffusion (without prior learning) contains *1065190* trainable parameters, and the proposed StochDiff (with prior learning) contains *1098214* trainable parameters. The difference in parameter count between these structures is only 33,024 parameters, representing a *3.1%* increase, which is negligible in terms of the network's overall capacity. Thus, we can conclude that the performance decline observed in the ablation study cannot be attributed to a significant reduction in parameter count. Given this fact, we believe the performance improvement observed with StochDiff is primarily due to the inclusion of the prior knowledge learning procedure, which enhances the model's ability to learn stochastic time series data.
> > >
> > > (2). The main contribution of our paper **does not lie in the design of fast sampling diffusion model**. Which is another important research domain in diffusion based models and we have also included in our **future perspective** part that a lightweight diffusion model with better fast sampling strategy is a valuable future work. Our main contribution, as stated in introduction, lies in the (1) proposal of new diffusion based architecture which include diffusion module during the time series modelling stage rather than the existing post-hoc design, (2) proposal of prior knowledge learning for diffusion module to enhance the model's learning ability on stochastic time series data, (3) showcase the model's real-world implication, highlighting its practical usage for real-world scenario. Thus, following common practice to showcase the model's applicability should not be considered as a drawback of our work.
> > >
> > > (3). It seems the question regarding the prior knowledge learning process has been addressed, and the concern remains on the motivation of this prior knowledge learning. Here, we would like to re-iterate our motivation regarding the data-driven prior learning: As we stated in the introdcution, the common setting of the generative models (e.g. VAE, Diffusion, etc.) is using fixed normal distribution $\mathcal{N}(0,1)$ as the latent prior, we argued that **this unimodal distribution exhibits limited capability to encode the full temporal dynamics and uncertainty of real-world time series data, considering these real-world data are mostly complex and multi-modal**. Our concern aligns with observations from prior work in natural language processing and computer vision domains, where researchers have highlighted similar limitations of fixed priors [1,2,3,4]. Experimental results of the existing diffusion based time series forecasting models also exhibit limitations of normal diffusion process on the stochastic time series data. Thus, we believe our proposal to learn a data-driven prior knowledge for diffusion module during the stochastic time series modelling is well motivated.
> > >
> > > References:
> > >
> > > [1] F. P. Casale, et al. Gaussian Process Prior Variational Autoencoders. 2018.
> > >
> > > [2] F. Lavda, Improving VAE generations of multimodal data through data-dependent conditional priors. 2019.
> > >
> > > [3] E. Aksan and O. Hilliges. STCN: Stochastic temporal convolutional networks. 2019.
> > >
> > > [4] S. Lee, et al. Priorgrad: Improving conditional denoising diffusion models with data-dependent adaptive prior. 2022.

---

> > > > ### Comment · Reviewer_aF5F · 2024-11-22
> > > >
> > > > Thanks for your further response.
> > > >
> > > > I have raised my score and recommend incorporating the key points from your response to question 1 into the paper. Additionally, I suggest revising the relevant sections (about sampling time) of the paper based on your response to question 2 to address potential sources of confusion for readers.

---

> > > > > ### Author Response · Authors · 2024-11-25
> > > > > **paper update**
> > > > >
> > > > > We would like to thank the reviewer to acknowledge our responses and provide valuable suggestions to further improve our work!
> > > > >
> > > > > We have now updated our paper based on these recommendations/suggestions (updated part are highlighted with blue font):
> > > > >
> > > > > (1). We added the details of parameter count in ablation study (section 5.4) to provide more insights of the model design and justify the architecture’s solidity. Additionally, as we mentioned, we have also updated the descriptions of our model (section C) to avoid any confusion on the prior learning procedure.
> > > > >
> > > > > (2). We added the detailed acceleration methods in sampling time part (section 5.5) to avoid any confusion on the contributions of our work.
> > > > >
> > > > > We hope the updated paper now meets the standards for acceptance from the reviewer's perspective. We are also willing to address any additional concerns the reviewer may have.

---

### Meta-Review · Area_Chair_Xm91 · 2024-12-20

**Metareview:**

The authors propose a method for generative time series forecasting that uses diffusion processes. The novelty of the paper includes the use of diffusion processes at each time step during modeling. Two of the reviewers thought the paper could potentially be accepted, while two other reviewers were in general more confident in their vote for rejection. All four reviewers found the contribution of the paper to be only "fair". Overall, the sentiment of the reviewers, both positive and negative, was that the contribution was incremental.

**Additional Comments On Reviewer Discussion:**

Three reviewers responded during the discussion phase. One chose to increase their score from 3 to 5, but did not change the general negative sentiment of the review. Overall, the reviewers appreciated the feedback, but did not seem enthusiastic about acceptance in its current form.

---

### Decision · Program_Chairs · 2025-01-22

Reject